# Pattern transfer of large-scale thin membranes with controllable self-delamination interface for integrated functional systems

Jun Kyu Park[1], Yue Zhang[2], Baoxing Xu [2] & Seok Kim [1,3,4✉]

Direct transfer of pre-patterned device-grade nano-to-microscale materials highly benefits many existing and potential, high performance, heterogeneously integrated functional systems over conventional lithography-based microfabrication. We present, in combined theory and experiment, a self-delamination-driven pattern transfer of a single crystalline silicon thin membrane via well-controlled interfacial design in liquid media. This pattern transfer allows the usage of an intermediate or mediator substrate where both front and back sides of a thin membrane are capable of being integrated with standard lithographical processing, thereby achieving deterministic assembly of the thin membrane into a multi-functional system. Implementations of these capabilities are demonstrated in broad variety of applications ranging from electronics to microelectromechanical systems, wetting and filtration, and metamaterials.

[1] Department of Mechanical Science and Engineering, University of Illinois at Urbana-Champaign, Urbana, IL 61801, USA. [2] Department of Mechanical and Aerospace Engineering, University of Virginia, Charlottesville, VA 22903, USA. [3] Institute for Convergence Research and Education in Advanced Technology, Yonsei University, Seoul 03722, South Korea. [4] Department of Mechanical Engineering, Pohang University of Science and Technology (POSTECH), Pohang 37673, South Korea. ✉email: seok.kim@postech.ac.kr

Pattern transfer on a substrate is essential for the integration of nano-to-microscale materials into functional structures and devices for a wide scope of applications. For example, lithographical pattern transfer (e.g. photolithography[1–3], e-beam lithography[4–6], imprinting lithography[7–9], etc.) that forms a photoresist pattern on a substrate has been ubiquitously utilized in top-down monolithic microfabrication together with successive process steps (e.g., etching and deposition). However, it suffers from the process specific drawbacks, such as the requirement of a flat substrate and the limited material compatibility. Alternatively, direct pattern transfer via transfer printing[10–15] and contact transfer[16–20] is relatively free from those challenges since it generates a pattern on a target substrate by conveying formerly patterned materials that are produced on a separate mother substrate. To date, these methods to transfer patterned materials on one substrate to another have been done by direct contact with a target substrate[17–20] or by a polymeric mediator that is often either a spin-coated layer[16] or a reversible dry adhesive[10–13]. In many cases, patterned materials formed on a mother substrate after undercutting a underneath sacrificial layer are transferred to a target substrate by either way. In the case of transfer printing, patterned materials together with a polymeric mediator are peeled off from a mother substrate and placed onto a target substrate. Then the removal of the polymeric mediator finishes the transfer of patterned materials to the target substrate. Although, this type of direct pattern transfer has been a powerful protocol to deterministically assemble nano-to-microscale materials onto target substrates, the entire pattern size of a transferred material has been limited, particularly for device-grade crystalline materials (e.g. Si, GaAs, GaN, etc.) which are highly rigid and brittle. A thin large area brittle material is commonly prone to fracture during transfer due to strain mismatch with a target substrate or polymeric mediator[21–24]. Thus, direct pattern transfer of a large area device-grade material piece without physical damage is a significant challenge, which would otherwise enable more diverse cost-effective functional integrated structures and devices.

In this work, we report a pattern transfer method that is enabled by self-delamination of a thin membrane from a substrate via controlled interfacial force in liquid environments particularly to directly transfer a thin and large area patterned single crystalline silicon (Si) membrane onto nearly any type of target substrate. Remarkably, the Si membrane can be lithographically processed on mediator substrates several times and then, in a well controlled self-delamination manner, transferred

onto a final target substrate for functional system assembly, as depeicted in Fig. 1, which has not been shown elsewhere to our knowledge. The theoretical model is established to understand the transfer mechanism based on self-delamination in the liquid media and provides a quantitative guide to experimental demonstrations in great agreement. It is worthwhile to note that the theoretical model certainly ensures the versatility and robustness of this method to be readily extended for other membrane materials while Si membranes are primarily utilized in this work. The membrane–substrate adhesion is controllable upon material selection, and thus the adhesion can be high enough to allow a lithographical process on a membrane but weak enough to retrieve it from a substrate using an elastomer surface. This ability provides an opportunity to build an ideal patterned Si platelet array which can be deterministically assembled into function structures or devices on a target substrate using transfer printing. In addition, the reported pattern transfer method makes complex 3D Si structures possible via one-step transfer since a patterned Si membrane can be transferred on and adapted for a structured target substrate due to the low flexural rigidity of the membrane. As opposed to other existing direct pattern transfer methods, this method enables flip and transfer of a patterned membrane which grants the choice of whether an initially patterned membrane surface faces up or down after transfer. This capability allows for the multiple lithographical processes on both front and back sides of a thin large area Si membrane as shown in Fig. 1. Here, we introduce the typical procedure of the reported pattern transfer method and theoretically address how interfacial force between contacting surfaces changes to allow for the pattern transfer involving thin memrance self-delamination in different environments. Next, we demonstrate the versatility of this method with the microassembly of both single-side and double-side patterned Si platelets. Moreover, we show the hybrid microassembly of a light emitting diode (LED) circuit relying on transfer of a metal patterned Si membrane and surface tension-driven self-alignment. Finally, the simple and economic fabrication of challenging re-entrant structures is exhibited. Particularly, the fabricated re-entrant structures show omniphobicitiy and even advanced functionalities such as directional omniphobicity and selective permeability for filtration applications. In addition to using Si, we also used polyimide (PI) and designed an auxetic patterned PI membrane with negative Poisson ratio to present the concept of stretch-induced tunable filtration.

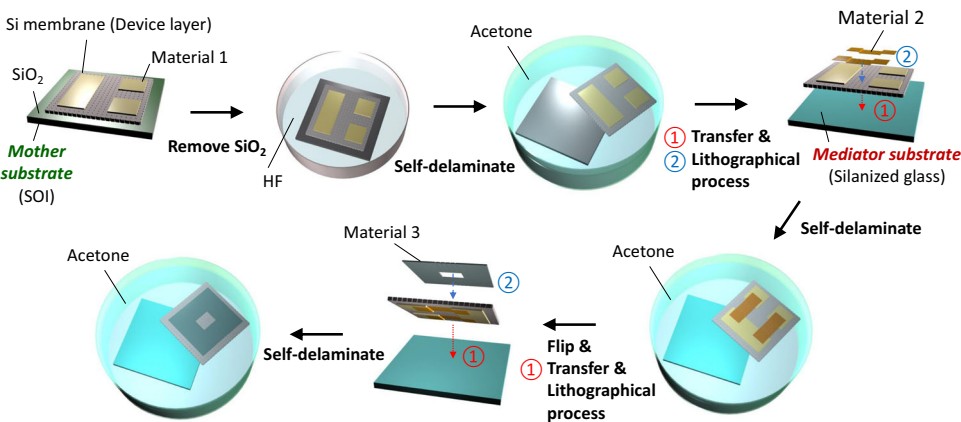

**Fig. 1 Self-delamination based pattern transfer and its implementation into classic lithographical process of functional devices.** See Supplementary Fig. 1 for the details of other variants. It starts with the initial patterning of a Si membrane on a mother substrate before transferring to a mediator substrate for subsequent processes. The processed Si memberane is flipped and transferred to another mediator substrate for further processing on the back side. Finally, the patterned Si membrane is self-delaminated from the mediator substrate and transferred to a target substrate for use.

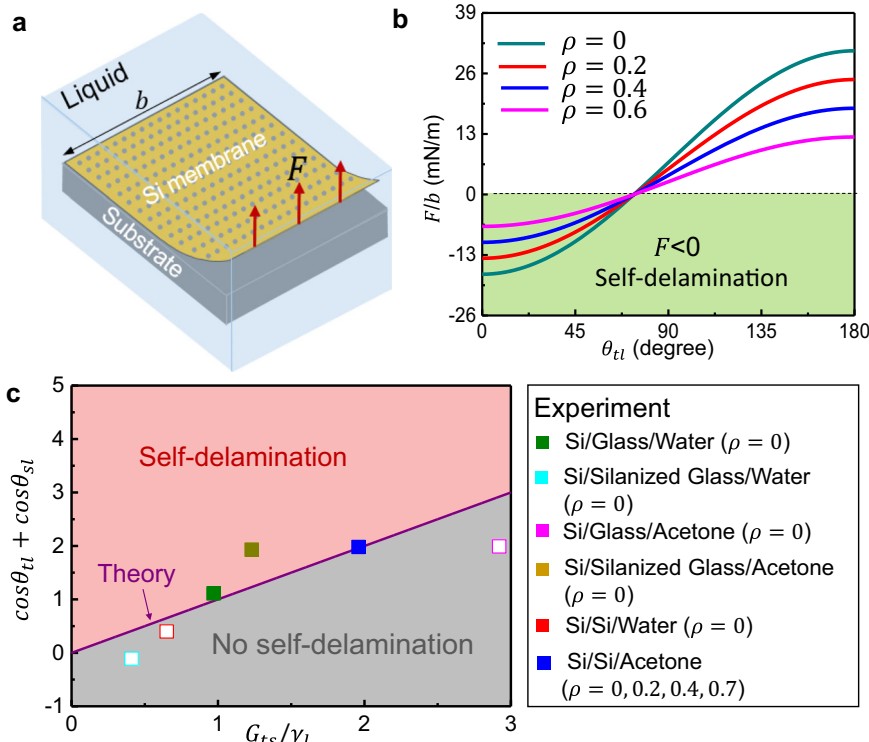

**Fig. 2 Theoretical model of thin film self-delamination from a substrate in liquid. a** Schematic illustration of mechanics model of peeling a thin film with arrays of microscale holes and width $b$ from a substrate in liquid by a peeling force $F$ at a 90° degree peeling angle. **b** Theoretical calculations of peeling force per unit width $F/b$ as a function of the thin film surface wettability $\theta_{tl}$ for films with different $\rho$ (total holes area over total film area), where $G_{ts} = 29.4$ mN/m, $\theta_{sl} = 20°$, and $\gamma_l = 24$ mN/m. **c** Theoretical phase diagram on the successful conditions of thin film self-delamination, which are experimentally confirmed on a wide variety of system materials for substrate, thin film, and liquid.

## Results and discussion

The reported pattern transfer method introduced in Fig. 1 is implemented in two different modes and the details are summarized in Supplementary Fig. 1. The procedure starts with the initial patterning of a Si device layer of a silicon-on-insulator (SOI) mother substrate with HF compatible material (material 1) and forming etch holes on it. Removal of SiO$_2$ layer in a HF bath makes the Si device layer stick on the mother substrate. Submerging it in an acetone bath enables the controlled interfacial force-driven self-delamination of the Si device layer, i.e., Si membrane, from the mother substrate. This is one example of the theoretically designed self-delamination of a membrane from a substrate in a liquid environment which is the key to the following patten transfer routes. For application demonstration of the self-delaminated Si membrane, it is transferred to other substrate in one of two following modes. In Mode 1, the Si membrane transferred on a target substrate is etched into small platelets using material 1 etch mask and the platelets are ready for use, which is labeled (1) in Supplementary Fig. 1. A variant of Mode 1 where the Si membrane transferred on a mediator substrate is lithographically processed with material 2 is labeled (2) in Supplementary Fig. 1. The processed Si membrane may further be transferred on the second mediator substrate after flip to process the back side surface with material 3. The Si membrane is finally transferred on a target substrate for use. In Mode 2, the initially patterned Si membrane is transferred on a target substrate for use without further process.

The mechanics of the reported pattern transfer method involves the deterministic self-delamination of a Si membrane from a mother or mediator substrate via controlled interfacial force. A Si membrane is modeled as a thin film with width of $b$ peeled off from a substrate by a peeling force $F$ at a 90° peeling angle in a liquid environment (Fig. 2a). For a quasi-static peeling process, with a small peeling distance $\triangle l$ in the direction of peeling force $F$, the energy balance between the work done by peeling force $\triangle W^F$ and the change of associated surface energy $\triangle E_{surface}$ at the steady-state transfer leads to $W^F = \triangle E_{surface} = F\triangle l$. When a porous film with porosity $\rho$ delaminates from a substrate in liquid, the change of effective contact area is $\triangle l(1 - \rho)b$. Therefore, the change of surface energy is $\triangle E_{surface} = \left(G_{ts} - \gamma_l(\cos\theta_{tl} + \cos\theta_{sl})\right)\triangle l(1 - \rho)b$[25,26], where $\gamma_l$ is liquid surface tension, $\theta_{tl}$ and $\theta_{sl}$ are the contact angle of liquid on thin film and substrate, respectively, and reflects their surface wettability. $G_{ts}$ is the interfacial adhesion energy between thin film and substrate in a dry air condition, and $G_{ts} = \gamma_t + \gamma_s - \gamma_{ts}$ where $\gamma_t$ and $\gamma_s$ are the surface tension of thin film and substrate, respectively, and $\gamma_{ts}$ is the interfacial tension between thin film and substrate. With $W^F = \triangle E_{surface} = F\triangle l$, the peeling force per unit width is now written as

$$F/b = (G_{ts} - \gamma_l(\cos\theta_{tl} + \cos\theta_{sl}))(1 - \rho). \quad (1)$$

When the required peeling force per unit width $F/b \leq 0$, Eq. (1) shows that a film is self-delaminated from a substrate in liquid, and at $F/b > 0$, applying an external force $F$ becomes necessary for achieving the delamination. Figure 2b represents the plot of the required $F/b$ along with the porosity and wettability of thin film when $G_{ts} = 29.4$ mN/m, $\theta_{sl} = 20°$, and $\gamma_l = 24$ mN/m. Similarly, Supplementary Figs. 2a, Sb show the effect of wettability of substrate and interfacial adhesion energy between thin film and substrate on the required $F/b$, respectively. The required peeling strength $F/b$ increases with the increasing of interfacial

adhesion energy $G_{ts}$ and $F/b$ becomes larger than 0 for all the porosity of film when $G_{ts}$ is beyond a critical value such as the large van der Waals interaction force[27]. In this scenario, applying an external mechanical peeling force is required to assist the delamination at the interface between film and substrate.

As indicated in Eq. (1), a thin film is easily peeled off from a substrate in a liquid environment when the wettability is high, the film–substrate interfacial adhesion energy is low, or the porosity of a film is large. To prove this, various pairs of substrate and liquid for a Si membrane are experimentally investigated and the favorable pairs for peeling are found. As shown in Fig. 2c, the theoretical diagram calculated using Eq. (1) with $\rho = 0$ agrees well with the experiment results. In the theoretical diagram, the purple curve represents the theoretical prediction on critical condition of thin film self-delamination. The symbols represent the experimental results, where the open symbol denotes no self-delamination and the solid symbol denotes self-delamination. The colors of symbols define material types of film/substrate/liquid. Supplementary Table 1 provides the values of parameters ($G$, contact angle, surface tension) which are calculated using the harmonic mean equation[28–30]. Guided by the theoretical analysis in Eq. (1) and the related experimental results, an acetone medium is selected to effectively peel a Si membrane from a Si mother substrate and a silanized glass substrate is chosen as an mediator substrate in this work. In addition, the theoretically calculated effect of porosity on $F/b$ is experimentally proven. Si membranes with different porosity of 0.04, 0.2, 0.4, and 0.7 are prepared as shown in Supplementary Fig. 3 and the experimental results also qualitatively shows that a higher porosity Si membrane is more favorable for peeling off from a Si substrate in an acetone medium. The computational modeling of the self-delamination of a film on substrates with a broad variety of materials is similar to that of peeling a film from substrates under an applied mechanical force where molecular dynamics (MD) simulations can be employed[25].

Once the self-delaminated patterned thin membrane is obtained, it can be readily implemented into other processes (outlined in Supplementary Fig. 1) for fabricating integrated functional systems. Figure 3 summarizes how Mode I (1) of the reported pattern transfer enables the production of microscale device-grade Au-coated Si platelets on a target substrate. The method begins with preparing a Si membrane with the initial pattern of deposited material (e.g. metals) on a SOI substrate (Supplementary Fig. 4a). After the complete removal of $SiO_2$ sacrificial layer, the Si membrane is delaminated from a mother SOI substrate by soaking in the acetone bath (Supplementary Fig. 4b). The Si membrane floating on liquid is transferred to a glass target substrate with liquid (Fig. 3a left). Then drying of the underlying liquid forms tight contact between two surfaces via surface tension-induced pressure $\Delta P = \gamma_l(\cos\theta_{tl} + \cos\theta_{sl})/h$, where $h$ is the distance between the Si membrane and the glass substrate[31,32]. After that, the Si membrane is dissected into an array of microscale Au/Si platelets by reactive ion etch (RIE) with the Au pattern as a hard mask (Fig. 3a, b). Consequently, a pack of aligned Au/Si platelets in various configurations regardless of the target substrate material (e.g. glass, PDMS, etc.) can be produced. A rose mosaic shape pattern on a glass substrate is shown in Fig. 3b. Additional patterns of Au/Si platelets on glass as well as curved PDMS substrates are shown in Supplementary Fig. 5. Apparently, the soft PDMS surface with a Au/Si platelet array is highly bendable, which envisions its potential applications toward flexible electronics.

Remarkably, the glass target substrate where a Si membrane is transferred and patterned may become a perfect mediator substrate if it is coated with an anti-stick monolayer or silanized. Here, the mediator substrate is with moderate adhesion not only

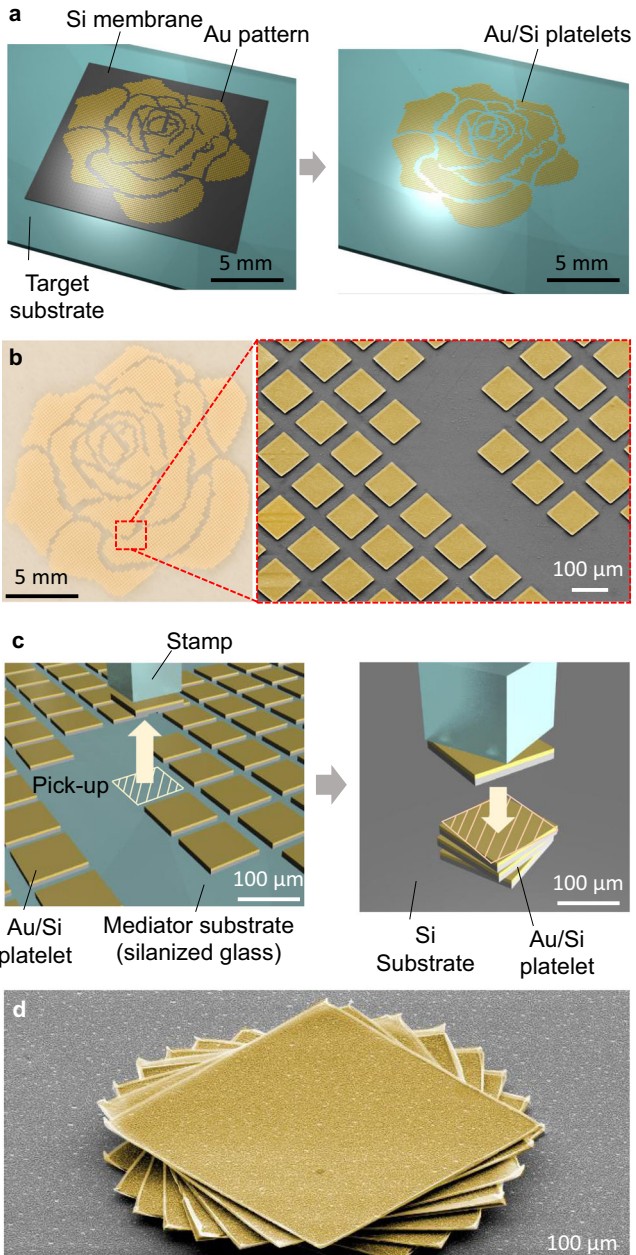

**Fig. 3 Au/Si platelets on a target substrate after Mode I (1) procedure in Supplementary Fig. 1 and their application to microassembly via transfer printing. a** A schematic illustration of a Au patterned Si membrane in a rose mosaic shape on a target substrate before and after Si etching. **b** Optical and magnified SEM images of the Au/Si platelet array. Au areas are yellow colored. **c** Cartoons of the microassembly of Au/Si platelets via transfer printing. **d** A colored SEM image of assembled Au/Si platelets with incremental rotations after thermal joining. Au areas are yellow colored.

to hold Au/Si platelets but also to ensure the reliable retrieval of them by a polymeric stamp. Figure 3c shows this trait with cartoons that retrieved Au/Si platelets are easily stacked without failure. Furthermore, these stacked Si platelets are thermally joined by employing the eutectic bonding between Si and Au surfaces to form a robust microscale structure with a unique 3D shape as shown in Fig. 3d[33,34].

Mode I (2) of the reported pattern transfer allows a Si membrane to be attached to and detached from a mediator substrate several times in liquid, which provides a powerful route to

multiple microfabrication processes of a Si membrane as depicted in the second row of Supplementary Fig. 1. Since the mediator substrate needs no sacrificial layer, there is no process constraint caused by using a sacrificial layer. For example, making a Si membrane directly from an SOI wafer does not allow depositing any materials incompatible with HF since it should finally be used to remove a $SiO_2$ sacrificial layer at the end of conventional Si membrane preparation protocols[12,18,33–35]. In addition, via Mode I (2), a Si membrane is easily flipped in liquid and stay upside-down on a mediator substrate. Therefore, both front and back sides of the Si membrane can be processed and patterned to have dual functionalities that do not interrupt each other. This ability is exceptional since conventional Si membrane preparation protocols require sophisticated patterning steps on only one side to have multi-functions, which often sacrifices other structural performance[12,18,33–35].

To demonstrate this exceptional ability, a double-side patterned sunflower-like Si platelet is fabricated as shown in Fig. 4a. On the front side, black Si and thin gold surfaces are patterned to mimic a sunflower using structural coloration and material color. On the back side, a NdFeB hard magnet material is deposited and magnetized in one direction. The detailed process steps are available in Supplementary Fig. 7. Then the double-side patterned Si platelet is retrieved and assembled on top of a flexible PDMS pillar, which finishes the fabrication of a sunflower-like structure with dual functionalities as shown in Fig. 4a. Due to the structural coloration, the center portion of the Si platelet is deep black. On the other hand, the assembled structure is actuated by an external magnetic field thanks to the strong magnetization in a NdFeB layer on the back side of the Si platelet. Implementing both the structural coloration by black Si and the magnetic motion by a NdFeB layer on only one side would be impossible as a NdFeB magnetic layer on top of black Si would diminish the structural coloration. Exceptionally, the reported Mode I (2) route to double-side processing of a Si platelet allows two incompatible functionalities to reside in a single Si platelet without compromising any performance.

The tilting angle of the sunflower-like structure under an external magnetic field $B$ can be predicted by equating the magnetic torque $V_mMB\sin(\pi/2-\theta)$ and the elastic restoring torque $K_{eq}\theta$, where $V_m$ is the volume of a magnetic material, $M$ is a magnetization strength, $K_{eq}$ is the equivalent torsion spring constant, and $\theta$ is the tilting angle[35]. As depicted in Fig. 4b, experimentally measured tilting angles match well with analytical computed tilting angles, which confirms the intact magnetic motion of the Si platelet. A series of optical images of the sunflower-like structure under different magnetic field strength (0, 0.55, 0.7, 0.8 T) is available in Supplementary Fig. 8. Finally, Fig. 4c shows the motion of the sunflower-like structure that tilts upon an external magnetic field mimicking sunflower motion that tracks the sun during the daytime.

Alternatively, a pre-patterned Si membrane can be directly transferred onto a target substrate where it is utilized as the final form, which is Mode II as depicted in the third row of Supplementary Fig. 1. Figure 5a shows a Au patterned Si membrane transferred on a target substrate with a disconnected red-green-blue (RGB) LED circuit. The optical images of a Si membrane as well as a target substrate are in Supplementary Fig. 9. Once a Si membrane is delaminated from a mother substrate, the Si membrane is flipped inside a liquid medium and transferred onto a target substrate. For the precision assembly, the target substrate has a hydrophilic pattern on a square region to induce the surface tension-driven self-alignment of the Si membrane during the evaporation of underlying deionized (DI) water as shown in Figs. 5b and S9b[36–38]. The actual self-alignment procedure is captured in Supplementary Movie 1. After the assembly, the

electric circuit is connected and then, serially positioned RGB LEDs are turned on with an external power supply. In addition, Fig. 5c, d show that the interaction between a Si membrane and a curved target substrate transforms a 2D patterned Si membrane into a 3D structure due to the low flexural rigidity of the Si membrane. For this purpose, a Si membrane should be thin and patterned into a shape that ensures the geometric flexibility (Supplementary Fig. 10a). After a Si membrane is transferred onto a 3D dome-shape PDMS target substrate, the strong contact in between is made during the evaporation of underlying liquid. The optical images and finite-element analysis (FEA) plots of 3D assembled Si membranes in saddle and dome shapes depending on their initial pattern designs are shown in Figs. 5d and S10b.

Mode II of the reported pattern transfer also varies to allow a pre-patterned membrane to be used on a target substrate for other novel functionalities. Figure 6 captures representative examples of omniphobic surfaces which are built using the pattern transfer Mode II. While omniphobic surfaces have received much attention due to their repellency toward liquids with extremely low surface tensions[39–43], forming them requires complex fabrication steps since they are commonly with re-entrant structural designs. The pattern transfer of this work may provide the cost effective route to these complex re-entrant structures. Figure 6a shows a pre-patterned Si membrane that is transferred on a continuous uncrosslinked SU8 negative photo-resist layer. Using an additional photomask combined with the transferred Si membrane enabling the self-aligned photo-lithography, only SU8 under rectangular openings is cured and a hexagonal lattice re-entrant structure is simply constructed. This structure is called Si/SU8 hereafter. Even simpler approach involves just a hexagonal patterned Si membrane transferred on a PDMS slab with a punched hole as depicted in Fig. 6b. The figure shows the transfer illustrations and the image of the hexagonal patterned Si membrane on the punched PDMS slab. This structure is called Si/PDMS hereafter. On these two re-entrant structures, even liquid with a low energy can build up an upward surface tension to be suspended as shown in Fig. 6c. To confirm their omniphobic characteristics, four liquids with different surface tensions ($\gamma$) including water ($\gamma = 72.8$ mN/m), ethylene glycol ($\gamma = 48.0$ mN/m), acetone ($\gamma = 24$ mN/m), and hexane ($\gamma = 18$ mN/m) are used. In addition, three different solid fraction hexagonal pattern designs of Si membranes and nanostructured counterparts are prepared for Si/SU8 and Si/PDMS surfaces. The geometric designs of the patterned Si membranes are in Supplementary Fig. 11. As shown in Fig. 6d, e, all surfaces are able to repel relatively high surface tension liquid such as water and ethylene glycol. However, only Si/PDMS surfaces can reliably suspend liquids with very low surface tensions such as acetone and hexane. For Si/SU8 surfaces with the solid fraction of 0.18, those low surface tension liquids smear into the structures and forms nearly 0° contact angle showing the Wenzel state wetting as shown in Fig. 6e. The trend between apparent contact angle ($\theta^*$) and solid fraction ($f$) is theoretically predicted using the equation of $\cos\theta^* = f(\cos\theta_Y + 1) - 1$ when droplets are in suspended state[39], where $\theta_Y$ is the intrinsic contact angle of water ($\theta_Y = 110°$), ethylene glycol ($\theta_Y = 85°$), acetone ($\theta_Y = 25°$), or hexane ($\theta_Y = 5°$) on a silanized either Si or SU8 surface. The solid fraction ($f$) of assembled surfaces are available in Supplementary Fig. 12. As shown in Fig. 6d, measured apparent contact angle values match well with theoretically calculated ones, which demonstrates the deterministic omniphobic characteristics of the re-entrant surfaces which are simply fabricated using the reported pattern transfer.

When patterned Si membranes are covered with black Si nanostructures, the roughness ratio ($r_f$) is increased to 2.8 from 1

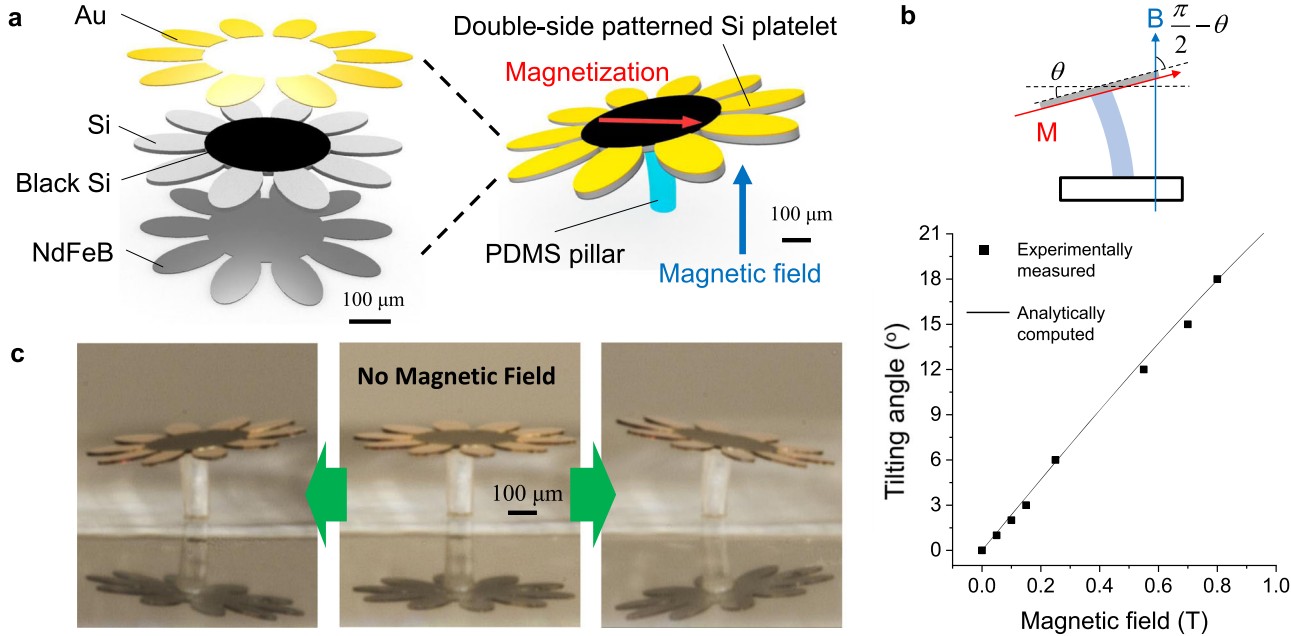

**Fig. 4 A double-side patterned Si platelet with dual functions of structural coloration and magnetic motion after Mode I (2) procedure in Supplementary Fig. 1. a** A schematic illustration of the composition of a double-side patterned Si platelet. One side is formed by black silicon nanostructures and gold patterns representing a sunflower and the other side is loaded with a NdFeB magnet alloy. The Si platelet transferred on a PDMS pillar is actuated by an external magnetic field after magnetization. **b** A graph showing tilting angle of the fabricated structure as a function of magnetic field strength. **c** Optical images for the tilting motion of the sunflower-like structure upon an external magnetic field.

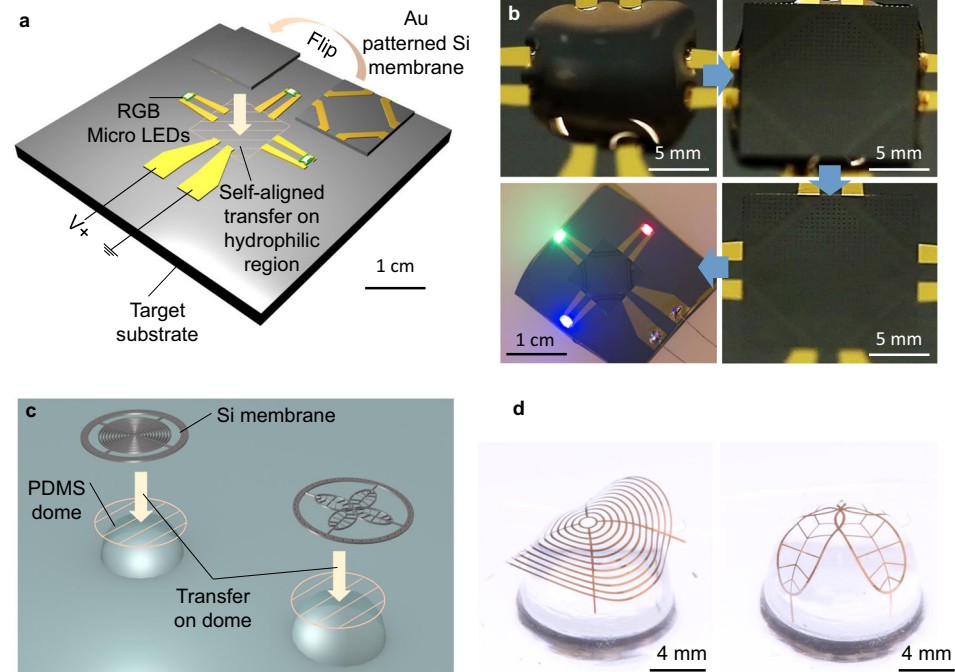

**Fig. 5 An LED circuit and single crystalline Si 3D mesostructures after Mode II procedure in Supplementary Fig. 1. a** A schematic illustration of the assembly process including flipping and transferring of a Au patterned Si membrane. A target substrate has a square hydrophilic region to enable the surface tension-driven self-alignment assembly. **b** A series of optical images of the self-alignment during underlying liquid drying and an optical image of assembled RGB LED circuit. **c** A schematic illustration of the process to build 3D Si mesostructures. Patterned Si membranes are transferred on structured PDMS domes. **d** Optical images of the resulted 3D Si mesostructures.

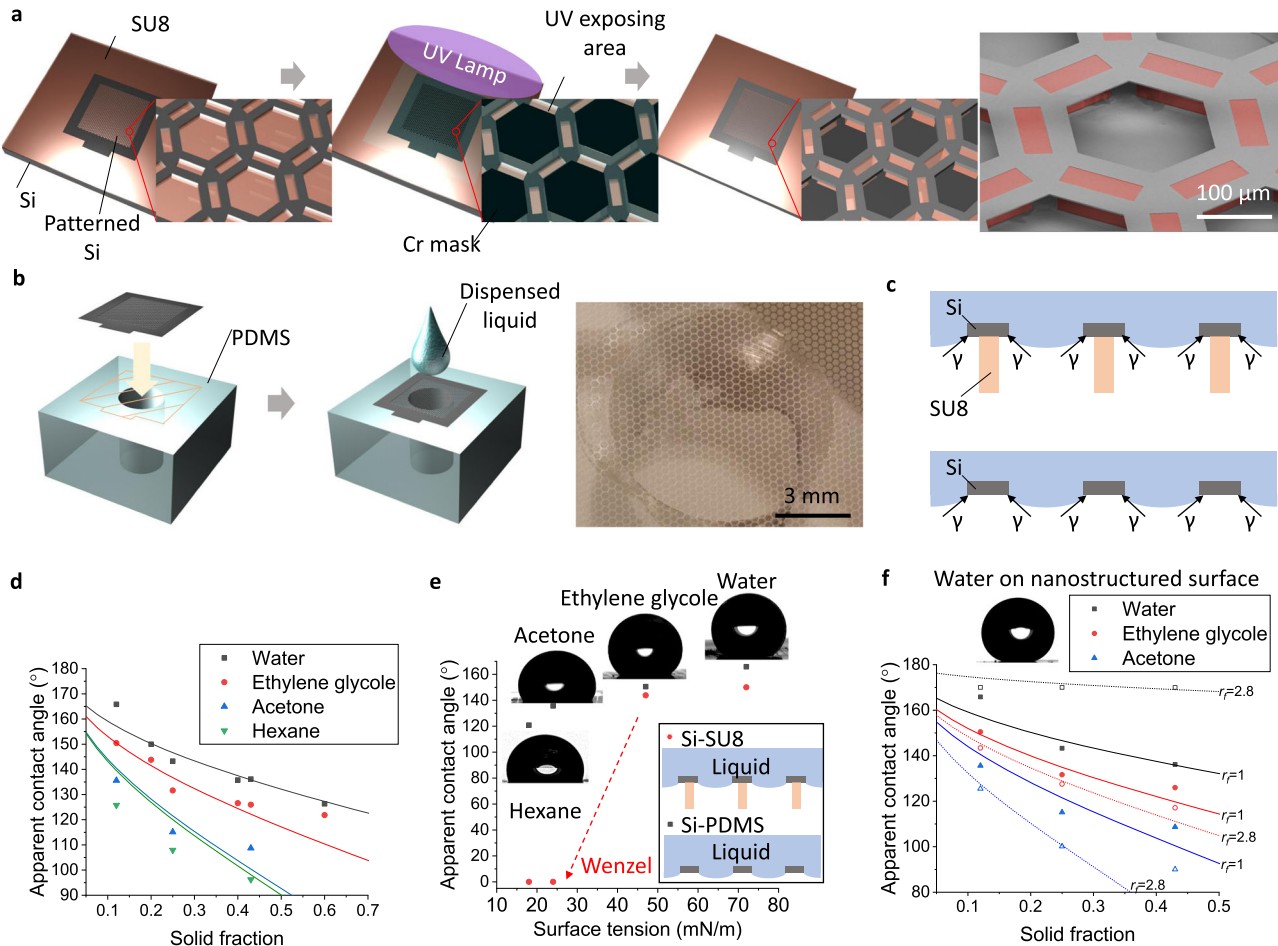

**Fig. 6 Patterned Si membranes demonstrating omniphobicity. a** A schematic step to fabricate a re-entrant surface composed of a pre-patterned Si membrane on an SU8 layer involving the self-aligned photolithography. The right SEM image with brown colored SU8 structures presents the high structural integrity of the resultant Si/SU8 surface. **b** A schematic step to form a hexagonal patterned Si membrane on a punched PDMS. The right optical image depicts the resultant Si/PDMS surface. **c** Illustrations showing the upward surface tension of a low energy liquid droplet placed on Si/SU8 (upper) and Si/PDMS (lower) surfaces enabling omniphobicity. **d** A graph showing apparent contact angle of diverse liquids as a function of different solid fraction of Si/SU8 and Si/PDMS surfaces. **e** A graph showing apparent contact angle of diverse liquids as a function of their surface tension on Si/SU8 (red) and Si/PDMS (black) surfaces with $f = 0.18$ and $f = 0.13$, respectively. **f** A graph showing apparent contact angle of diverse liquids as a function of solid fraction of Si/PDMS surfaces. Si surfaces are either smooth ($r_f = 1$, solid symbol) or nanostructured ($r_f = 2.8$, hollow symbol).

and it makes the assembled Si/PDMS surfaces superhydrophobic as shown in Fig. 6f. The roughness ratio of the nanostructured surface is measured using an AFM technique and is found in Supplementary Fig. 13. However, the roughness ratio acts differently for liquids with lower surface tension ($\theta_Y < 90°$), and causes the reduced omniphobicity. This difference is explained using the equation of $\cos\theta^* = f\left(r_f \cos\theta_Y + 1\right) - 1$ that is used for a solid surface with a roughness[39]. Apparently, the negative $\cos\theta_Y$ term for a lower surface tension ($\theta_Y < 90°$) liquid decreases $\cos\theta^*$. Figure 6f does not include data for hexane which is challenging for the nanostructured Si/PDMS surface to reliably suspend.

Furthermore, inspired by the simple Si/PDMS surface fabrication step presented in Fig. 6, a mechanical metamaterial made of PI is designed to show the stretch-induced switchable wettability. Here, auxetic (metamaterial sample) and hexagonal (control sample) patterned PI membranes are transferred and bonded to a punched PDMS slab. The fabrication steps for PI membranes are available in Supplementary Fig. 14. When the hexagonal lattice PI membrane is uniaxially stretched, the lattice dimension ($L_m$) normal to the stretching direction decreases because of its positive Poisson's ratio (Fig. 7a). However, that of

the auxetic lattice PI membrane increases because its Poisson's ratio is negative. The ability to suspend a liquid droplet is inversely proportional to the lattice dimension ($L_m$). With the negative Poisson's ratio, the mechanical stretch-induced switchable omniphobicity is achieved in the auxetic lattice PI membrane as opposed to conventional grid pattern surfaces. To show this capability, a droplet of vegetable oil is placed and suspended on both hexagonal and auxetic lattice PI membranes. Then the membranes are uniaxially stretched and only the oil droplet on the auxetic lattice PI membrane penetrates it. The detailed experimental results are captured in Supplementary Movie 2.

Remarkably, the assembled Si/PDMS surface shown in Fig. 6 demonstrates the switchable wettability macroscopically depending on the configuration of the assembly although its microscopic wettability is still omniphobic. When a Si membrane is facing upward as depicted in left column of Fig. 7b, the macroscopic wettability is dictated by the property of PDMS (hydrophobic and oleophilic) as a liquid droplet meets with the underneath punched PDMS sidewall. Here, a 500 μm-thick PI sheet is added to ensure the structural rigidity of the assembled surface. Therefore, vegetable oil ($\gamma = 32$ mN/m) pass through the assembled surface. On the other hand, when the Si membrane is

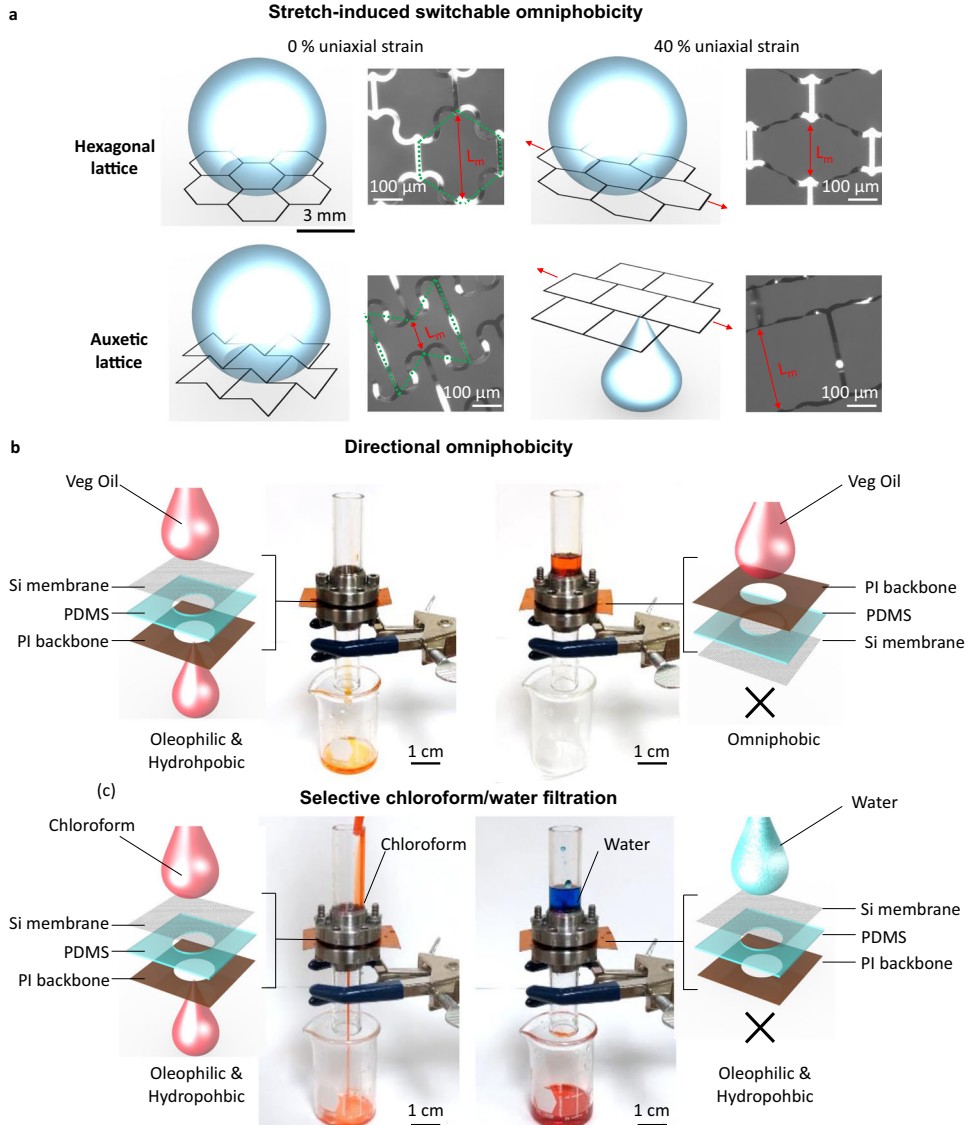

**Fig. 7 PI membranes showing stretch-induced switchable omniphobicity, and hexagonal patterned Si membranes demonstrating directional omniphobicity and selective filtration. a** Illustrations and images of PI membranes with auxetic lattice design for novel stretch-induced switchable wettability and with counterpart hexagonal lattice design. **b** An assembled oleophilic Si/PDMS/PI surface where vegetable oil passes (left) and an omniphobic PI/PDMS/Si surface where vegetable oil does not pass (right). **c** Selective filtration of chloroform/water mixture using the assembled oleophilic yet hydrophobic Si/PDMS/PI surface where chloroform passes, yet water is trapped.

facing down as depicted in right column of Fig. 7b, the assembled surface becomes purely omniphobic as liquid does not meet any other surfaces under the Si membrane. Consequently, vegetable oil does not pass through the assembled surface. More demonstrative experimental results are available in Supplementary Movie 3.

The straightforward application of these unique capabilities of the assembled Si/PDMS/PI surface should be to separate water from low surface tension liquids. As a part of example, the mixture between water ($\gamma = 72.8$ mN/m) and chloroform ($\gamma = 27$ mN/m) is prepared and poured into the assembled surface where a Si membrane is facing upward as shown in Fig. 7c. Since the macroscopic wettability of the assembled surface is oleophilic and hydrophobic, chloroform passes through the assembled surface and is collected in an underneath container while water is trapped on top of the assembled surface. Supplementary Movie 4 presents more details of the experimental results. Since the macroscale wettability of the assembled surface

is controlled by the material underneath the patterned Si membrane, other variations are also possible by changing the underneath material. Supplementary Fig. 15 and Supplementary Movie 5 depict one of the variations where the assembled surface is omniphilic using a water and oil permeable material, such as TexWipe® under the Si membrane. While the assembled Si/PDMS surface show promising potentials towards directional wettability and resultant filtration applications, its functionalities are from unique structural designs enabled by the simple pattern transfer methods presented in this work.

This work reports the unique capabilities of a thin membrane pattern transfer method which relies on determinate control of interfacial force between contacting surfaces in liquid environments and theoretical explanations to understand key parameters determining optimal membrane-substrate–liquid combinations. Particularly, a pre-patterned device-grade single crystalline Si membrane is easily transferred to nearly any type of substrates encompassing a silanized glass mediator substrate for use or

further microfabrication processes. Upon these unique capabilities, the versatility of this approach is summarized by demonstrating the microassembly of single- and double-side patterned Si membranes, the hybrid assembly of RGB LED circuit especially assisted with the surface tension-driven self-alignment. Moreover, re-entrant shape patterned surfaces are straightforwardly and cost-effectively fabricated for surface directional omniphobicity and selective filtration purposes, which highlights the potential of the reported approach towards diverse applications. Future opportunities include extending thin membrane material choices possibly using guidance from further theoretical models and enabling more precision assembly processes employing advanced equipment and tooling.

## Methods

**Fabrication and transfer of Si membrane with Au pattern**. Fabrication of a Si membrane starts with rinsing an SOI wafer (3 μm-thick device layer and 1 μm-thick SiO$_2$ sacrificial layer) with acetone, IPA, and DI water followed by drying under a stream of nitrogen. Cr (5 nm)/Au (50 nm) is deposited using e-beam evaporation (FC-2000, Termescal) onto the device layer of SOI wafer and wet etched through a mask of photoresist (AZ5214, Microchem) as shown in Supplementary Fig. 3a. Then the Si device layer is patterned into a desired shape using SF$_6$ (20 sccm) and O$_2$ (2 sccm) plasma at 150 W, 20 mTorr for 6 min. After that, the sample is immersed in HF bath for 6–12 h to remove the SiO$_2$ sacrificial layer. The patterned Si membrane is delaminated from the mother SOI substrate by immersing in acetone bath as shown in Supplementary Fig. 4b. Then the floating Si membrane is scooped with acetone and transferred onto a target substrate. Drying of underlying liquid completes the pattern transfer of a Si membrane with Au pattern.

**Patterning of Si membrane into Au/Si platelets and transfer printing of Au/Si platelets**. Once a Si membrane with Au pattern is transferred on a target substrate. The Si membrane is dissected into Au/Si platelets using SF$_6$ (20 sccm) and O$_2$ (2 sccm) plasma at 150 W, 20 mTorr for 6 min with the Au pattern as an etching mask as shown in Supplementary Fig. 6. When the target substrate is a silanized glass substrate, Au/Si platelets on it are easily picked up using a PDMS stamp. The PDMS stamp is brought into contact with a Si platelet with preload and rapidly retracted to pick up a Au/Si platelet. The Au/Si platelets are transferred to another target Si substrate with translation and angular alignment to build a stacked structure shown in Fig. 3d. After stacking, the Au/Si platelets are thermally joined in rapid thermal processing (RTP) furnace at 360 °C for 10 min.

**Fabrication of sunflower mimicking structure**. A device layer of an SOI wafer (10 μm-thick device layer and 1 μm-thick SiO$_2$ sacrificial layer) is partially patterned with black Si nanostructures. To pattern black Si nanostructure, native oxide on Si surface is removed in HF bath for 1 min. Then thin oxide layer was grown by O$_2$ plasma at 10 sccm, RF1 120 W, RF2 200 W at 50 mTorr for 5 min. The oxide layer is incompletely etched by CHF$_3$ plasma at 12 sccm, 350 W at 50 mTorr for 2 min. Using the left oxide islands as an etching mask Si is slowly etched by Cl$_2$ (40 sccm) and Ar (4 sccm) plasma RF1 250 W, RF2 300 W at 90 mTorr for 10 min to form sharp and dense nanostructures. Then the Si membrane is patterned by DRIE process such that a Si platelet array is mechanically supported by the Si membrane (Supplementary Fig. 7a). Then the Si membrane is delaminated and flipped from the mother substrate inside the liquid medium, and transferred on a silanized glass substrate. On the glass substrate, Cr (5 nm)/Au (50 nm) is patterned by a lift-off process with a photoresist masking layer (SPR220, Microchem). Once the front side process is finished, the Si membrane is delaminated again in acetone bath (Supplementary Fig. 7b) and transferred to a new silanized glass substrate for backside processing as depicted in Supplementary Fig. 7c. On the Si membrane, Ti (10 nm)/NdFeB (400 nm)/Ti (10 nm) are sputter deposited. After the deposition, the NdFeB layer is magnetized under 1.9 T magnetic field and delaminated from the glass substrate (Supplementary Fig. 7d). After transfer to another new silanized glass substrate, a Si platelet is mechanically untethered form the Si membrane and picked up by a PDMS pillar to be a sunflower mimicking structure (Supplementary Fig. 7e, 7f).

**Fabrication of LED circuit and single crystalline Si 3D mesostructures**. To build a LED circuit, Cr (5 nm)/Au (20 nm) is deposited by e-beam evaporation (FC-2000, Temescal) on a SOI device layer (3 μm) and a Si wafer. Then the deposited layers are wet etched through a mask of photoresist (AZ5214, Microchem) as shown in Supplementary Fig. 9. To fabricate a Si membrane, a gold-patterned SOI wafer device layer is etched by SF$_6$ (20 sccm) and O$_2$ (2 sccm) plasma with 150 W at 20 mTorr for 6 min. To pattern a square hydrophilic region on a target substrate, a mask of photoresist is patterned on the square area and silane (Trichloro(1H,1H,2H,2H,-perfluorooctyl)silane, Sigma Aldrich) is selectively deposited using the photoresist as a masking layer (Supplementary Fig. 9b). The target substrate is cleaned with acetone and IPA to remove the masking layer. Then

the Si membrane with Au/Cr pattern (Supplementary Fig. 9a) is flipped inside acetone bath and moved to DI water bath. After that, the Si membrane transferred onto the hydrophilic region with DI water. DI water is let dried for 24 h to ensure tight contact is made between the Si membrane and the target substrate. RGB LEDs (SMD 0603, Dialight) are connected to the assembled circuit using silver epoxy (8331S, MG chemicals). Finally, 10 V is applied to turn the LED lights on through soldered copper wire.

To build single crystalline Si 3D mesostructures, an SOI device layer (3 μm) is patterned by using SF$_6$ (20 sccm) and O$_2$ (2 sccm) plasma with 150 W at 20 mTorr for 6 min with photoresist as masking layer (Supplementary Fig. 10a). PDMS dome structures are prepared by pouring 10:1 base to curing agent ratio PDMS precursor (Sylgard 184, Dow Chemical) on a mold. Then the Si membranes are transferred onto the cured PDMS dome structures with underlying liquids. Here the outer ring in Supplementary Fig. 10a is used to manually align an Si membrane and a PDMS dome structure and is removed after contact.

**Fabrication of omniphobic Si/SU8 surfaces**. Negative photoresist (SU8-50, Microchem) is spun on a Si substrate with 3000 rpm and soft baked at 65 °C for 5 min and at 95 °C for 15 min. This yielded 40 μm-thick soft baked SU8 layer on a Si substrate. A separately prepared pre-patterned Si membrane is delaminated from its mother substrate in acetone bath and transferred to IPA bath for selectivity. Then it is transferred to the soft baked SU8 layer. Drying out the underlying IPA and subsequent heating on 65 °C for 5 min makes conformal contact between SU8 and the SI membrane. After that, 200 mJ/cm$^2$ of UV light is selectively exposed to rectangular openings, which defines supporting SU8 pillars connected to the Si substrate. The assembled substrate is hard baked at 65 °C for 1 min and 95 °C for 5 min. Using SU8 developer (Microchem), SU8 is developed to define rectangular pillar. The resultant structure surface is modified by depositing silane (Trichloro(1H,1H,2H,2H,-perfluorooctyl)silane, Sigma Aldrich).

**Fabrication of hexagonal and auxetic lattice PI membranes**. PMMA (PMMA A7, Microchem) is spun on a Si wafer at 3500 rpm for 40 s and baked at 180 °C for 3.5 min. Then PI (PI-2545, Dupont) is spun on the PMMA layer at 2000 rpm for 60 s and cured at 250 °C in a vacuum oven. This yields a 3 μm-thick PI layer with sufficient thickness uniformity. Cr (5 nm)/Au (20 nm) is deposited by e-beam evaporation (FC-2000, Temescal) onto the PI layer and wet etched through a mask of photoresist (AZ5214, Microchem) as the PI layer to have hexagonal or auxetic designs (Supplementary Fig. 14a). Using the Cr/Au layer as an etch mask, the PI and PMMA layers are dry-etched using an O$_2$ plasma at 20 sccm with 200 W at 50 mTorr for 10 min (Supplementary Fig. 14b). Then the sample is immersed in acetone bath for 24 h to remove the PMMA sacrificial layer and release the PI membrane from the wafer. The PI membranes are transferred onto punched PDMS slabs to suspend hexagonal or auxetic design parts (Supplementary Fig. 14c). Then the assembled surface is cut into a stretchable shape (Supplementary Fig. 14d).

## Data availability

All data used and generated in this study are available in Supporting Information.

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

## Acknowledgements

This work was financially supported by the National Science Foundation (Grant No. ECCS-1950009 for J.K.P. and S.K.) (Grant No. CMMI-1928788 for Y.Z. and B.X.).

## Author contributions

J.K.P. and S.K. conceived the idea. J.K.P. performed the experimental studies. J.K.P. and Y.Z. carried out the analysis. J.K.P. and Y.Z. wrote the manuscript. All authors read and revised the manuscript. S.K. and B.X. supervised the work.

## Competing interests

The authors declare no competing interests.
