## [Peer Review File · Nature Communications]

Pattern transfer of large-scale thin membranes with controllable self-delamination interface for integrated functional systemsREVIEWER COMMENTS

Reviewer #1 (Remarks to the Author):

The authors should be congratulated on an excellent paper, showing a novel fabrication method for membrane transfers. It is a timely and interesting paper. Aside from typos and some grammatical awkwardness, the paper should be published.

I would only make the following three comments:

1. it would be helpful to discuss general approaches which make this method more widely applicable, i.e. to different materials. In particular, a discussion on the theory would be helpful, as would a discussion on the theoretical approaches that could be used to predict the behaviour of the system/approach for various material pairs.
2. a discussion of van der Waals interactions and their impact on the feasibility of the approach. In particular, the authors may want to discuss the large interaction forces computed using many body approaches in Nature communications 11 (1), 1-8 (2020).
3. What would be the computational and modelling approaches required to generalise the approach proposed here, and how could this be done in a consistent manner, regardless of the material pair.

Reviewer #2 (Remarks to the Author):

The authors present an innovative fabrication technique that can achieve direct transfer of pre-patterned nano-to-microscale materials. The fabrication process is driven by self-delamination of single crystalline silicon thin membrane in liquid media. Theoretical model is established to help understand the key factors that affect the self-delamination and how to control the transfer process. This direct pattern transfer technique possesses some unique advantages compared with other methods. It can be used to generate complex 3D Si structures. This method also enables flip and transfer of patterned membrane, which allows for multiple lithographical process on both front and back sides. Then the versatility of this fabrication method is demonstrated through several applications, including hybrid microassembly of LED circuit and re-entrant structures fabrication with omniphobicity and other advanced functionalities.

The idea of utilizing thin film self-delamination to achieve pattern transfer is innovative. It provides a convenient method to achieve direct transfer of large area without physical damage. Designs in this manuscript for multifunctional applications are likely to be of broad interest, but there are a few points that need further clarification

Comments:

1. The authors provide multiple examples with patterns on different scales. How the performance/robustness of the technique varies in these scenarios? One factor particularly is the thickness of the thin films. Since the pattern transfer process happens in liquid, the surface tension or environmental perturbation may cause potential fractures, especially for patterns on nano scale. The examples provided in the manuscripts are mostly over 100 μ m. Can the authors comment on this?
2. The values of parameters for theoretical model in equation 1 are provided directly without any source or reference. How do the authors estimate the value of surface tension between thin film and substrate (γ_{ts})?
3. The key step of the fabrication technique is the controlled interfacial adhesion in liquid environments. The authors provided the theoretical model that explains the process in detail and matches the experimental results well, which can be used as guidance for other extensible applications. Eq. (1) suggests high wettability and low film-substrate adhesion energy is favorable,

but such requirements may also limit the application for this technique, especially for Mode I (2), where the pre-patterned Si membrane will be process more than once and multiple materials are involved. The complexity makes it more difficult to guarantee the delamination happens only in the film-substrate interface. The authors can provide more in-depth discussion on possible extensions to other materials/liquid environments for further applications of the technique.

4. The manuscript could have been better if more detailed analysis of this new technique can be provided. The applications of the fabrication methods take relatively large portion of the manuscript while some of them are not directly related to pattern transfer. In the example application provided in Fig. (7), stretch-induced switchable wettability is achieved by utilizing special patterned membranes, which is a well-studied topic. It is not clear how the self-delamination driven pattern transfer method contributes to the application. What are the unique advantages of this method compared with others in this application?

Reviewer #3 (Remarks to the Author):

The authors reported a thin membrane pattern transfer method by self-delamination via controlled interfacial force. The authors studied the mechanics of the pattern transfer method to understand the self-delamination of Si membrane from the substrates theoretically and experimentally. The authors demonstrated the assembly of single- and double-side patterned Si membranes and the hybrid assembly of a LED circuit. Also, they demonstrated re-entrant structures, showing omniphobicity and selective permeability to provide advanced filtration. This reviewer believe that the pattern transfer method is novel and would be useful for many applications. A couple of minor comments are shown below to improve the manuscript.

1. The authors selected acetone as a medium to effectively peel a Si membrane. Can the authors elaborate on more details of why the acetone was selected compared to other solvents such as IPA, ethanol, water, etc. Please also provide more details of why silanized glass substrate was chosen as a mediator substrate.

2. This reviewer agrees with the fact that the pattern transfer of thin membranes can be powerful method to assemble materials deterministically to target substrates regardless fragile materials. However, how can this method be cost-effective if it consumes the SOI wafer at every fabrication?

REVIEWER COMMENTS

Reviewer #1 (Remarks to the Author):

The authors should be congratulated on an excellent paper, showing a novel fabrication method for membrane transfers. It is a timely and interesting paper. Aside from typos and some grammatical awkwardness, the paper should be published.

I would only make the following three comments:

1. it would be helpful to discuss general approaches which make this method more widely applicable, i.e. to different materials. In particular, a discussion on the theory would be helpful, as would a discussion on the theoretical approaches that could be used to predict the behaviour of the system/approach for various material pairs.

Our response : We highly appreciate the reviewer's positive evaluation. We also sincerely thank him/her for the comment on the process applicability to wide range of materials. In fact, Si membrane/Si substrate pair is one of the hardest cases to self-delaminate as the Si/Si pair has very high interfacial adhesion energy G_{ts} . In addition, Si is rigid/brittle with easy fracture at very small deformation as well as Si/Si pair has flawless contact interface. Even for these challenging conditions, our approach can be widely applicable as long as the thin membrane material is not dissolved to medium liquid such as water/solvents and has decent sacrificial material pair. As a preliminary example, we also showed that pattern transfer of polyimide thin membrane/Si substrate pair with a PMMA sacrificial layer in Fig 7a. Certainly, we believe that this technique can further be extended into other device-grade materials such as an epitaxially grown GaAs membrane/Si or GaAs substrate pair with a AlGaAs sacrificial layer [R1]

Our modification to the manuscript: To clarify this point, we have modified and included the following sentence in Page 2 "The theoretical model is established to understand the transfer mechanism based on self-delamination in the liquid media and provides a quantitative guide to experimental demonstrations in great agreement. It is worthwhile to note that the theoretical model certainly ensures the versatility and robustness of this method to be readily extended for other membrane materials while Si membranes are primarily utilized in this work."

[R1] Park, Sang-II, et al. "Printed assemblies of inorganic light-emitting diodes for deformable and semitransparent displays." *science* 325.5943 (2009): 977-981.

2. a discussion of van der Waals interactions and their impact on the feasibility of the approach. In particular, the authors may want to discuss the large interaction forces computed using many body approaches in *Nature communications* 11 (1), 1-8 (2020).

Our response: We appreciate the reviewer for this valuable suggestion and sharing the important reference with us. The increasing of van der Waals interaction force will lead to the increasing of interfacial adhesion energy G_{ts} between film and substrate. As shown in Supplementary Fig. 2, the increasing of G_{ts} leads to the increasing of required mechanical

peeling strength F/b , which will make the self-delamination more difficult. For example, when the wettability of film and substrate is $\theta_{sl} = 20^\circ$ and $\theta_{tl} = 7.65^\circ$, and surface tension is $\gamma_l = 24\text{mN/m}$, the peeling strength F/b becomes larger than 0 when G_{ts} is larger than 46mN/m for all different porosity of film. Therefore, this self-delamination approach cannot be used when G_{ts} is larger than 46mN/m .

Our modification to the manuscript: To clarify this point, we have modified and included the following sentence in Page 4 “Similarly, Supplementary Fig. 2a and 2b show the effect of wettability of substrate and interfacial adhesion energy between thin film and substrate on the required F/b , respectively. The required peeling strength F/b increases with the increasing of interfacial adhesion energy G_{ts} and F/b becomes larger than 0 for all the porosity of film when G_{ts} is beyond a critical value such as the large van der Waals interaction force [27]. In this scenario, applying an external mechanical peeling force is required to assist the delamination at the interface between film and substrate.”

[27] Hauseux, P., Nguyen, T. T., Ambrosetti, A., Ruiz, K. S., Bordas, S. P., & Tkatchenko, A. (2020). From quantum to continuum mechanics in the delamination of atomically-thin layers from substrates. *Nature communications*, 11(1), 1-8.

3. What would be the computational and modelling approaches required to generalise the approach proposed here, and how could this be done in a consistent manner, regardless of the material pair.

Our response : We appreciate the reviewer for this valuable comment. In our previous work (Yue Zhang, Qingchang Liu, and Baoxing Xu. *Extreme Mechanics Letters* 16 (2017): 33-40), we have modeled the peeling of thin film from substrate by an applied mechanical peeling force in a liquid medium using molecular dynamics (MD) simulations for different material pair. The self-delamination approach proposed here is similar to that mechanical peeling method except that there is no external mechanical force, so the computational modeling of self-delamination approach can also be done using MD simulation in a consistent manner.

Our modification to the manuscript: To clarify this point, we have modified and included the following sentence in Page 5 “Si membranes with different porosity of 0.04, 0.2, 0.4, and 0.7 are prepared as shown in Supplementary Fig. 3 and the experimental results also qualitatively shows that a higher porosity Si membrane is more favorable for peeling off from a Si substrate in an acetone medium. The computational modeling of the self-delamination of a film on substrates with a broad variety of materials is similar to that of peeling a film from substrates under an applied mechanical force where molecular dynamics (MD) simulations can be employed [25].”

[25] Zhang, Y., Liu, Q., & Xu, B. (2017). Liquid-assisted, etching-free, mechanical peeling of 2D materials. *Extreme Mechanics Letters*, 16, 33-40.

Reviewer #2 (Remarks to the Author):

The authors present an innovative fabrication technique that can achieve direct transfer of pre-patterned nano-to-microscale materials. The fabrication process is driven by self-delamination of single crystalline silicon thin membrane in liquid media. Theoretical model is established to help understand the key factors that affect the self-delamination and how to control the transfer process. This direct pattern transfer technique possesses some unique advantages compared with other methods. It can be used to generate complex 3D Si structures. This method also enables flip and transfer of patterned membrane, which allows for multiple lithographical process on both front and back sides. Then the versatility of this fabrication method is demonstrated through several applications, including hybrid microassembly of LED circuit and re-entrant structures fabrication with omniphobicity and other advanced functionalities.

The idea of utilizing thin film self-delamination to achieve pattern transfer is innovative. It provides a convenient method to achieve direct transfer of large area without physical damage. Designs in this manuscript for multifunctional applications are likely to be of broad interest, but there are a few points that need further clarification

Comments:

1. The authors provide multiple examples with patterns on different scales. How the performance/robustness of the technique varies in these scenarios? One factor particularly is the thickness of the thin films. Since the pattern transfer process happens in liquid, the surface tension or environmental perturbation may cause potential fractures, especially for patterns on nano scale. The examples provided in the manuscripts are mostly over 100 μm . Can the authors comment on this?

Our response : We appreciate the reviewer's valuable comments on the robustness of the pattern transfer method. The key factor to maintain the integrity of the patterned thin film is to ensure the lateral stiffness of the film. When a thin film is too thin so the lateral stiffness of the thin film is too small, it might cause wrinkles on the transferred film depending on the drying methods [R2,R3]. This could be the drawback of the pattern transfer method but at the same time it could be an engineering opportunity toward the self-assembled surface profile.

On the other hand, the defined inner patterns like Supplementary Fig. 7a and Fig 5d are more susceptible to the surface tension as it has higher aspect ratio. The structures might lose their original shape during transfer and drying process. To prevent this, we introduced physical guide to hold the flexible structures to hold in place during the transfer and drying process. It is quite true that the inner structure become more susceptible to fracture in lower scale during wet transfer. Possible breakthrough is to have more physical guides or to pattern a thin film into the inner structure after transfer like Fig. 3a instead of before transfer.

[R2] Calado, V. E., et al. "Formation and control of wrinkles in graphene by the wedging transfer method." *Applied Physics Letters* 101.10 (2012): 103116.

[R3] Kim, Hyun Ho, et al. "Wetting-Assisted Crack-and Wrinkle-Free Transfer of Wafer-Scale Graphene onto Arbitrary Substrates over a Wide Range of Surface Energies." *Advanced Functional Materials* 26.13 (2016): 2070-2077.

2. The values of parameters for theoretical model in equation 1 are provided directly without any source or reference. How do the authors estimate the value of surface tension between thin film and substrate (γ_{ts})?

Our response : We appreciate the reviewer's valuable comments on how the surface tension between two solid surfaces could be estimated. In this manuscript, the interfacial energy between two materials (γ_{12}) is approximated by the harmonic mean equation [28-30].

$$\gamma_{12} = \gamma_1 + \gamma_2 - 4 \frac{\gamma_1^d \gamma_2^d}{\gamma_1^d + \gamma_2^d} - 4 \frac{\gamma_1^p \gamma_2^p}{\gamma_1^p + \gamma_2^p} \quad (\gamma = \gamma^d + \gamma^p)$$

Here, γ , γ^d , and γ^p are the surface tension, the dispersive and polar components of the surface tension of each material. The three parameters are obtained by measuring contact angles of two liquids (water and ethylene glycol) with known surface tensions on each material surfaces and using the below equation [28-30].

$$(1 + \cos \theta) \gamma_l = 4 \frac{\gamma_s^d \gamma_l^d}{\gamma_s^d + \gamma_l^d} + 4 \frac{\gamma_s^p \gamma_l^p}{\gamma_s^p + \gamma_l^p}$$

Where θ is the measured contact angle of a liquid on a surface, and γ_l , γ_l^d , and γ_l^p are the surface tension, and its polar and dispersive components of a liquid, respectively.

Our modification to the manuscript: To clarify this point, we have modified and included the following sentence in Page 4 “The colors of symbols define material types of film/substrate/liquid. Supplementary Table 1 provides the values of parameters (G, contact angle, surface tension) which are calculated using the harmonic mean equation [28-30].”

[28] Park, J. K., Eisenhaure, J. D., & Kim, S. (2019). Reversible underwater dry adhesion of a shape memory polymer. *Advanced Materials Interfaces*, 6(3), 1801542.

[29] Wu, S. (1971). Calculation of interfacial tension in polymer systems. In *Journal of Polymer Science Part C: Polymer Symposia* (Vol. 34, No. 1, pp. 19-30). New York: Wiley Subscription Services, Inc., A Wiley Company.

[30] Yeom, J., & Shannon, M. A. (2010). Detachment Lithography of Photosensitive Polymers: A Route to Fabricating Three-Dimensional Structures. *Advanced Functional Materials*, 20(2), 289-295.

3. The key step of the fabrication technique is the controlled interfacial adhesion in liquid environments. The authors provided the theoretical model that explains the process in detail and matches the experimental results well, which can be used as guidance for other extensible applications. Eq. (1) suggests high wettability and low film-substrate adhesion energy is favorable, but such requirements may also limit the application for this technique, especially for Mode I (2), where the pre-patterned Si membrane will be process more than once and multiple materials are involved. The complexity makes it more difficult to guarantee the delamination happens only in the film-substrate interface. The authors can provide more in-depth discussion on possible extensions to other materials/liquid environments for further applications of the technique.

Our response : We appreciate the reviewer’s concern on the possibly difficulty in delamination as the complexity involves. The theoretical model that we have provided assumed the flawless contact between thin film and substrate. In our perspective, the new material and etched-profile on top of thin-membrane surface introduces surface roughness which reduces the interaction between film-substrate rather than increasing the adhesion. Even in the worst case where the new material deposition increases the film-substrate adhesion, surface roughness can be manipulated on certain part of wafer to modulate the self-delamination. As an example of the further applications of this technique, we believe that this technique can be extended into other device-grade materials over silicon such as an epitaxially grown GaAs membrane/Si or GaAs substrate pair with a AlGaAs sacrificial layer [R1]. However, we think doing further experiments to prove these hypothesis and claim may be out of scope of this manuscript.

Our modification to the manuscript: To clarify this point, we have modified and included the following sentence in Page 2 “The theoretical model is established to understand the transfer mechanism based on self-delamination in the liquid media and provides a quantitative guide to experimental demonstrations in great agreement. **It is worthwhile to note that the theoretical model certainly ensures the versatility and robustness of this method to be readily extended for other membrane materials while Si membranes are primarily utilized in this work.**”

[R1] Park, Sang-Il, et al. "Printed assemblies of inorganic light-emitting diodes for deformable and semitransparent displays." *science* 325.5943 (2009): 977-981.

4. The manuscript could have been better if more detailed analysis of this new technique can be provided. The applications of the fabrication methods take relatively large portion of the manuscript while some of them are not directly related to pattern transfer. In the example application provided in Fig. (7), stretch-induced switchable wettability is achieved by utilizing special patterned membranes, which is a well-studied topic. It is not clear how the self-delamination driven pattern transfer method contributes to the application. What are the unique advantages of this method compared with others in this application?

Our response : We appreciate the reviewer’s valuable question on the application stated in Fig 7b. The uniqueness of this application is that it utilizes mechanical metamaterial ‘Auxetics’ to achieve switchable wettability. Although the stretch-induced switchable wettability is widely studied, the switchable wettability demonstration using the unique auxetic structures like our one seems to be quite rare or none, to our knowledge. Thus, we think that our unique pattern transfer method enabled the auxetic structure for stretch-induced switchable wettability that has not been shown before. Please forgive us if we missed any significant work on wetting control using auxetics and guide us with any related work for our further revising. Finally, this application as an example of Mode 2 uses polyimide thin film rather than silicon membrane. This means the expandability of pattern transfer material choice to the other material over silicon. That is another reason why we have Fig. 7a in this work.

Reviewer #3 (Remarks to the Author):

The authors reported a thin membrane pattern transfer method by self-delamination via controlled interfacial force. The authors studied the mechanics of the pattern transfer method to understand the self-delamination of Si membrane from the substrates theoretically and experimentally. The authors demonstrated the assembly of single- and double-side patterned Si membranes and the hybrid assembly of a LED circuit. Also, they demonstrated re-entrant structures, showing omniphobicity and selective permeability to provide advanced filtration. This reviewer believe that the pattern transfer method is novel and would be useful for many applications. A couple of minor comments are shown below to improve the manuscript.

1. The authors selected acetone as a medium to effectively peel a Si membrane. Can the authors elaborate on more details of why the acetone was selected compared to other solvents such as IPA, ethanol, water, etc. Please also provide more details of why silanized glass substrate was chosen as a mediator substrate.

Our response : We appreciate the reviewer's valuable comments on how we chose acetone and silanized glass substrate over other combination. The condition favorable to self-delamination happens (negative F/b) when we have low G_{ts} or $\gamma_{substrate}$ and low θ_{tl} and θ_{sl} in equation (1). The surface energy $\gamma_{substrate}$ of the silanized glass is the lowest among the substrate candidates available to us. In addition, the low surface tension of acetone among our own candidates (lower than or similar to IPA, ethanol, and water surface tension) allows to have low contact angle θ_{tl} and θ_{sl} . This is why we chose the acetone/silanized glass combination over the other candidates.

Our modification to the manuscript: To clarify this point, we have modified one sentence to the following sentence in Page 4 "Guided by the theoretical analysis in Equation (1) and the related experimental results, an acetone medium is selected to effectively peel a Si membrane from a Si mother substrate and a silanized glass substrate is chosen as an mediator substrate in this work."

2. This reviewer agrees with the fact that the pattern transfer of thin membranes can be powerful method to assemble materials deterministically to target substrates regardless fragile materials. However, how can this method be cost-effective if it consumes the SOI wafer at every fabrication?

Our response : We appreciate the reviewer's valuable comments on the cost-effectiveness of this process. In this method an SOI wafer is segmented to a number of functional pieces and each piece is functionalized to a device. In this aspect, the cost of introduced process can be divided by the number of potential devices that can be created from an SOI wafer. On top of that, we showed that material candidates of this process can be expanded by presenting a polyimide thin film transfer in Fig. 7a. However, the transfer process is low-cost, in particular, the self-delamination process on a mediator substrate does not sacrifice any materials as it is driven by physical interactions at solid-liquid interface. Therefore, we believe the pattern transfer of thin membranes can be cost-effective although the SOI does cost.

REVIEWERS' COMMENTS

Reviewer #2 (Remarks to the Author):

The authors have addressed all my comments in the revision. I have no further comments.

Reviewer #3 (Remarks to the Author):

The authors thoroughly addressed all of the reviewers' comments which I believe the manuscript is in a good shape to be published in this journal as it is.

REVIEWER COMMENTS

Reviewer #1 (Remarks to the Author): None.

Reviewer #2 (Remarks to the Author): The authors have addressed all my comments in the revision. I have no further comments.

Reviewer #3 (Remarks to the Author): The authors thoroughly addressed all of the reviewers' comments which I believe the manuscript is in a good shape to be published in this journal as it is.

Our response : We highly appreciate the reviewer's positive evaluation.